# What factors influence innovation efficiency in integrating digitalization and low carbonization within the construction industry? A configuration analysis based on fsQCA

**Shiming Wang, Haifeng Xiong** [ID]*

School of Business Administration, Liaoning Technical University, Huludao, China

* 2110192898@qq.com

## Abstract

In the dual context of the digital age and the 'double carbon' objectives, enhancing the innovation efficiency of integrating digitalization and low carbonization in the construction industry has become an inevitable trend. This study utilizes the fuzzy set Qualitative Comparative Analysis (fsQCA) method, framed within the Technology-Organization-Environment (TOE) theoretical perspective. By analyzing data from 30 provinces, the research examines how technological, organizational, and environmental factors influence the innovation efficiency of integrating digitalization and low carbonization in the construction sector from a configurational standpoint. The findings reveal that, as a whole, there is a necessary condition for the potential absence of high innovation efficiency in this integration within the construction industry. Additionally, no singular necessary condition was identified that affects high innovation efficiency in the sector. The study identifies four equivalent configuration pathways to enhance innovation efficiency by integrating digitalization and low carbonization. These pathways suggest that provinces can select a trajectory more suitable for the synergistic advancement of "digitalization and low carbonization" in the construction sector based on local city conditions, ultimately achieving the "dual carbon" goal. The research findings support the "Porter hypothesis," highlighting the critical role of environmental regulation in improving the innovation efficiency of this integration within the construction industry.

## Introduction

In 2020, during the 75th session of the United Nations General Assembly, China pledged to reach peak carbon dioxide emissions by 2030 and achieve carbon neutrality by 2060. This commitment underscores China's role in the international effort to combat climate change and acts as a catalyst for domestic action. It showcases China's dedication to global climate responsibility and enhances its status as a major power that is responsible. The Nation's "14th Five-Year Plan" explicitly emphasizes the development of emerging technologies, such as 5G

**Data availability statement:** All relevant data are within the manuscript and its Supporting Information files.

**Funding:** This work was supported in part by the Basic Scientific Research Project of EducationDepartment of Liaoning Province under grant LJKMR20220709 and in part by the Social Science Fund of Liaoning Province under grant 23-A045. The funders had no role in study design, data collection and analysis, decision to publish, or preparation of the manuscript.

**Competing interests:** The authors have declared that no competing interests exist.

and big data centers, within the construction and industrial sectors. Research indicates that the digital era significantly incentivizes the growth of low-carbon industries [1]. The construction sector is a major contributor to carbon emissions. According to the 2023 China Building Energy Consumption and Carbon Emission Research Report, the national building process consumed 2.35 billion tons of coal equivalent (tce). It produced 5.01 billion tons of carbon dioxide (tCO$_2$) in 2021. This accounts for 44.7% and 47.1% of the country's total energy consumption and carbon emissions, respectively, with annual increases of 4.5% and 4.3%. These figures highlight the crucial role of the construction industry in achieving the "double carbon" objectives. As the primary source of carbon emissions, the success of emission reduction strategies in this sector will significantly influence China's overall rate of carbon emissions reduction and the timeline for reaching its peak. Considering the ongoing year-on-year growth, it is likely that carbon emissions from the construction industry will continue to increase without effective interventions, posing challenges to meeting the "double carbon" targets. To achieve the "dual-carbon" goal, the construction industry must undergo significant innovation and transformation. This transformation will require integrating digital technology to facilitate a shift towards low-carbon, environmentally conscious, and sustainable development.

However, several challenges must be addressed to integrate digitalization and low carbonization into the construction industry. Firstly, although the domestic construction industry in China has a substantial market size, the digital construction market remains relatively small. There is a lack of investment in research and development for digital science and technology, leading to a digitalization level that is considerably below international standards. Moreover, in critical digital tools like Building Information Modeling (BIM) software, domestic products often lack competitiveness and need to enhance capabilities for independent and controlled operations. Secondly, the current distribution of digital talent in China is mainly concentrated in sectors such as the Internet, information, and communication technology, leaving the construction industry with a paucity of skilled IT professionals. There is a lack of organizational structures, clear career paths, and competency assessment standards tailored to the digitalization needs of the construction sector. Thirdly, while the government has implemented various policies and measures to support the advancement of the construction engineering sector, there is still a significant gap in specific policy guidance needed for the coordinated development of digitalization and low-carbonization. Addressing these challenges is essential for the construction industry to effectively contribute to achieving the "dual-carbon" objective. What are the optimal strategies for developing a "waste-free city," promoting the synergistic advancement of industrial digitization and low carbonization, and integrating digitization and low carbonization in an efficient and innovative manner within the construction industry? This is a genuine issue that requires immediate resolution and is also a matter of scientific concern to the academic community.

Significant research has been conducted by academics on digitalization and low carbonization. Numerous scholars have concentrated on the manufacturing sector, investigating how digital innovation influences the green development of manufacturing enterprises [2]. Researchers frequently examine whether digitalization can facilitate the green and low-carbon development of enterprises by analyzing various components such as the digital economy [3], digital finance [4], digital industry [5], and digital infrastructure [6]. In the construction industry, many scholars concentrate on the application of BIM technology in green building design [7] and its use throughout the entire life cycle of green buildings [8]. Additionally, some scholars investigate the pathways and strategies for the construction industry's green transformation within the intelligent construction framework [9,10].

However, a limited amount of research addressing the efficiency of integrating digitalization and low-carbon innovation in the construction industry. The existing literature typically

employs traditional regression models to examine the independent effects of technological factors on enterprises' green and low-carbon efficiency. This approach often neglects the complexity and multifactorial nature of integrating digitalization and low-carbon innovation within the construction sector. To effectively understand and address the causal complexity of such outcomes, it is more advantageous to analyze them from a configurational perspective [11–13]. Therefore, this article employs fuzzy set qualitative comparative analysis (fsQCA) to explore the sufficient configurational pathways that lead to the efficient integration of digitalization and low-carbon innovation in the construction industry. It also investigates whether any single condition might be a necessary factor influencing this integration efficiency. Beyond technological factors, the actions and attitudes of governmental organizations in various regions, along with environmental regulations and other environmental factors, are critical drivers of the digital and low-carbon transformation in the construction sector. The Technology-Organization-Environment (TOE) theoretical framework effectively identifies the antecedent conditions of complex research issues [14]. Therefore, this article selects antecedent conditions influencing the outcomes based on the existing theoretical framework of research—the TOE framework. This study aims to address the following questions: (1) Do the five antecedent conditions identified in the TOE framework—namely, technological innovation level, technical talent, government financial support, government promotion, and environmental regulation—function as necessary conditions that lead to high efficiency in the integration of digitalization and low carbonization within the construction industry? (2) Can these five antecedent conditions—technological innovation level, technical talent, governmental financial support, governmental publicity and promotion, and environmental regulation—be configured into several grouping paths that sufficiently result in high efficiency for digitalization and low carbonization integration in the construction industry? (3) Are there grouping paths formed by these five antecedent conditions—technological innovation level, technical talent, governmental financial support, governmental publicity and promotion, and environmental regulation—that lead to non-high innovation efficiency in integrating digitalization and low carbonization within the construction industry?

## Literature review

### The digitisation of the construction industry

Academic research on digital transformation in construction enterprises remains theoretical mainly, focusing primarily on three key aspects. First, it addresses the challenges and uncertainties associated with digital transformation in the construction industry. Compared to high-tech sectors, the construction industry exhibits a lower level of acceptance and implementation of advanced digital technologies [15]. Additionally, the industry's profitability tends to be relatively low, further complicating digital adoption [16]. The inherent characteristics of the construction industry present significant challenges to the implementing digital transformation through advanced digital technologies. To address these challenges, evaluating the application of specific digital tools within the sector is crucial. A survey conducted among construction professionals in Singapore revealed that while the majority of respondents recognize the potential of integrated digital delivery technologies to improve project performance, the adoption rate remains relatively low [17]. Berlak et al. conducted case studies on the German construction industry, examining the use of production robots, 3D printing, and Building Information Modeling (BIM) software. Their findings indicate that these digital technologies significantly enhance productivity within the sector [18]. Additionally, domestic scholars have investigated the advantages and feasibility of utilizing Internet of Things (IoT) technology [19] and virtual construction technology in project management [20]. Thirdly, it

is essential to examine digital transformation's evolution and development trends from the Building 3.0 era to the Building 4.0 era. According to Lee et al., the future will witness further advancements in the design and development of IoT, big data, additive manufacturing, 3D printing, and blockchain-driven BIM construction services [21]. Three emerging visions for the digital transformation of the architecture industry include the development of efficient buildings, creating user data-driven building environments, and implementing value-driven computational design [22].

## The low carbonization of the construction industry

As a major contributor to energy consumption and carbon emissions, the construction industry has increasingly prioritized low-carbon development, making it a focal point and prominent topic in current research. Scholars worldwide have explored various aspects, including concepts, technologies, and driving factors. For instance, Gocuk et al. [23] assert that Building Information Modeling (BIM) technology holds significant potential for reducing costs in architectural design firms and enhancing the quality of low-carbon buildings. Similarly, Li Xiao's research highlights BIM technology's substantial value and broad prospects in achieving comprehensive life cycle management of buildings. This technology advances the scientific and technological standards of planning, design, construction, and operation within the construction industry, thereby fostering low-carbon development in China's construction sector [8]. Dai Linbo emphasizes the necessity of fully utilizing modern information technology to manage green, low-carbon engineering projects. By doing so, engineering projects' green performance and resource utilization efficiency can be enhanced, contributing significantly to global low-carbon development [24]. Meanwhile, Xu Junmin observes that BIM has yet to be widely and effectively applied to buildings focusing on green and sustainable development. However, it holds substantial potential to support and improve environmental sustainability analysis, monitoring, Management, and optimization across the entire building life cycle [25]. Wang et al. conducted practical research on the impact of technological innovation on greenhouse gas emissions and concluded that advancements in construction technology adversely affect carbon dioxide emissions[26]. Similarly, Dowd et al. investigated the responsiveness of building envelope structures through dynamic thermal performance simulation methods, highlighting the role of these structures in facilitating the transition towards low-carbon practices in the construction sector [27]. Further, Zhao Xinyao explored the driving factors behind low-carbon transformation within construction enterprises, identifying factors such as the policy environment and technological innovation capabilities as positively impacting this transformation [28].

Existing research indicates that factors such as policy and technology play crucial roles in the digitalization and low-carbon transformation of the construction industry. This evidence reinforces the rationality of the antecedent conditions selected for analysis in this paper.

## Synergies between digitization and low carbonization in the construction industry

Yue Qingrui asserts that accelerating the deep integration of digital technology with green construction practices is a crucial pathway for the low-carbon transformation of the construction industry [29]. Albin Karlsson's research suggests that construction companies must develop a digital strategy to navigate the low-carbon digital transformation successfully [30]. Guo Hongling et al. contend that, within the context of the "dual carbon" goals, intelligent low-carbon construction emerges as the primary direction and objective for the industry's transformation and upgrading [31]. Wang Jiangying and colleagues emphasize that, as a

significant contributor to carbon emissions, the construction industry has a responsibility to utilize digital technology as a foundational tool. They advocate for the seamless integration of prefabricated building technology, green low-carbon technology, BIM technology, and intelligent operation and maintenance technology across the entire lifecycle of buildings. This approach aims to achieve the goals of "greening, industrialization, and informatization," thereby promoting industry transformation and upgrading towards high-quality development [32]. Further, research by Wang Bo et al. highlights the strategic importance of blending modern information technology with construction technology. This integration is vital for the green and low-carbon transformation of the industry, enhancing its development quality and helping to achieve China's "dual carbon" objectives [10].

## Analysis of the current research status

Despite the body of research on digital technology in the construction industry, several shortcomings persist.

Firstly, there is a notable gap in the research concerning the integration and innovative fusion of digitalization and low carbonization, especially regarding the key factors that impact the efficiency of this integrated innovation. While prior studies have mainly highlighted the benefits and potential of digital technologies in advancing green and low-carbon practices, they often overlook how these processes can synergistically work together. There is also a lack of research identifying factors that could significantly enhance the efficiency of this integration. This gap limits our understanding of how digital transformation and low carbonization can mutually reinforce each other in the construction industry.

Secondly, from a theoretical standpoint, there is an absence of a complete and systematic analytical framework to explore the efficiency of digitalization and low-carbonization innovation integration in the construction industry. This lack of a unified and scientific theoretical tool impedes the accurate assessment of the impact of digital technologies on the industry's low-carbon transformation. It limits the analysis of potential obstacles and opportunities in the innovation process. Such a framework is crucial for advancing research and providing effective guidance for practical applications.

Finally, in practice, the integration and innovative efficiency of digitalization and low-carbonization in the construction industry is a multifaceted process influenced by various factors such as government policies, environmental considerations, and technological advancements. These elements are interrelated, forming a complex web of causal relationships. However, existing research often fails to account for this complexity, neglecting to thoroughly explore the intrinsic connections and interaction mechanisms among these factors. As a result, there are ongoing challenges in effectively identifying and managing the complexities introduced by these factors in pursuing of integrated and innovative development of digitalization and low-carbonization in the construction industry.

## Model building

This article employs the fsQCA method to address the above shortcomings to analyze relevant data from 30 provinces (autonomous regions and municipalities). It explores the interactive effects of various antecedent conditions on the efficiency of integrating digitalization and low carbonization innovation in the construction industry within the TOE theoretical framework. A theoretical model influencing the efficiency of integrating digitalization and low carbonization innovation in the construction industry has been constructed.

First, there are technical conditions, including technological innovation, and two secondary conditions related to technical talent. On the one hand, digitalization has provided a

strategic positioning for the low carbonization transformation of the construction industry [29], especially as BIM technology can act as a driver for digital transformation in the construction environment [33]. However, due to limitations in technological innovation, there is a lack of interoperability between BIM and other tools [34] and a lack of mature cross-system linkage control systems for construction equipment [32]. On the other hand, people are fundamental; without technical talent, there can be no new technologies. Currently, our country has a serious shortage of composite technical talent and inadequate policies for talent training, guidance, and introduction [9,35]. The government should strengthen the training of technical talent and focus on attracting skilled professionals, thereby ensuring that technical conditions effectively contribute to the efficiency of innovation in integrating digitalization and low carbonization in the construction industry, achieving the "dual carbon" goals.

The second aspect involves organizational conditions, encompassing government financial support and the government's role in promoting the integration of digitalization and low-carbon innovation concepts within the construction industry. As a key driver of low-carbon transformation, government intervention significantly influences the outcomes of integrating these innovations in the sector. According to resource dependence theory, organizations must engage with others who control critical resources to thrive [36]. The government is the "visible hand" in resource allocation [37]. The government acts as the "visible hand" in resource allocation. On the economic front, financial support from the government is essential for encouraging construction enterprises to engage in digital and low-carbon integration initiatives [38]. Such support can lower barriers to implementation and incentivize companies to invest in innovative technologies and processes. However, a "crowding-out effect exists," where companies might misuse national subsidies for non-integrated innovations, potentially leading to adverse effects [39]. Beyond financial incentives, the transformation and innovation within the construction industry require a shift in traditional corporate mindsets toward embracing digital and low-carbon principles. The government plays a crucial role in fostering this mindset shift by actively promoting these concepts. Organizing lectures, observation meetings, and competitions specifically focused on digital and low-carbon innovations can enhance public understanding of such buildings and cultivate broader societal awareness. Increased publicity and training can mobilize participation and oversight from various societal sectors [40]. Therefore, this article zeroes in on how organizational financial support and conceptual promotion impact the efficiency of integrated innovation in the construction industry's digitalization and low-carbon transformation. The study aims to provide insights into the mechanisms that can effectively drive sustainable development and innovation within the sector by examining these aspects.

Thirdly, there are environmental conditions, which include environmental regulation as a secondary condition. The environmental regulation system is the most important policy framework within China's formal environmental management system [41]. There is no consensus on the relationship between environmental regulation and innovation in the construction industry. Some scholars believe that environmental regulation helps improve the efficiency of integrating digitalization and low carbonization innovation [41,42]. However, other scholars disagree, arguing that environmental regulation does not have a significant promotional effect on corporate transformation and innovation [43] and that it may even have adverse effects due to the "crowding out effect" of capital [44].

Based on the above analysis, this article finally selects five antecedent conditions: the level of technological innovation and technical talent in technical conditions, government financial support, and the promotion of the concept of integrating digitalization and low carbonization innovation in the construction industry by the government in organizational conditions, and

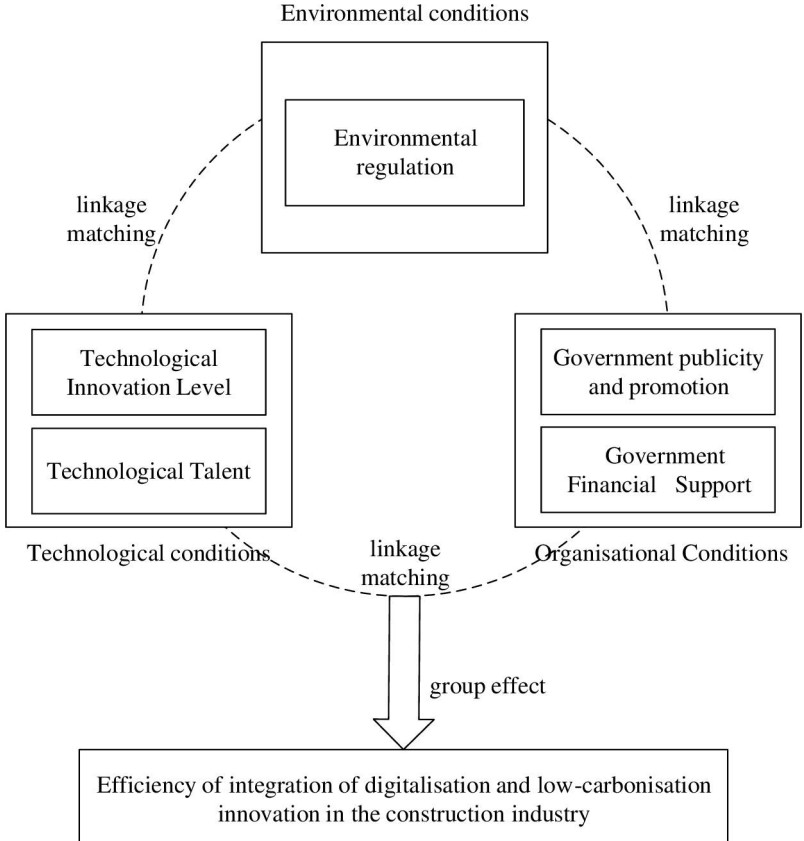

**Fig 1. Theoretical model framework.**

environmental regulations in environmental conditions. The theoretical research framework constructed is shown in Fig 1.

## Research methods and data processing

### Qualitative comparative analysis methods

Qualitative Comparative Analysis (QCA) is a method proposed by sociologist Charles Ragin based on set theory and Boolean operations. The QCA method posits that a single factor does not lead to an outcome or that a single element cannot sufficiently lead to an outcome; it is generally believed that the result is produced by the concurrent influence of multiple factors [12].

This article abandons the traditional symmetry research methods and chooses the fuzzy set qualitative comparative analysis (fsQCA) method from QCA for the following reasons: traditional regression models typically assume that the relationships between variables are symmetric and linear. However, in integrating digitalization and low carbonization in the construction industry, the influence of multiple antecedent conditions is not a linear relationship. fsQCA can handle these nonlinear and complex interactions. Analyzing the changes in outcomes under different combinations of interacting factors reveals the nonlinear relationships between them, thus providing a more accurate and comprehensive explanation. In addition, fsQCA employs a multi-case study approach, which can fully consider the heterogeneity and complexity of cases across different provinces. It is based on set theory

and configurational thinking, examining the relationship between antecedent conditions and outcomes from the perspective of sets. By establishing a multivariate analysis of causal relationships, fsQCA can identify the configurations of antecedent conditions that influence the integration of digitalization and low carbonization innovation in the construction industry and explore the equivalence between different antecedent configurations. fsQCA allows variables to take multiple continuous values between 0 and 1, meaning that the variables are no longer "black or white," but have different degrees. This application of fuzzy sets enables fsQCA to more accurately reflect the ambiguity and uncertainty in the construction industry's digitalization and low carbonization process.

## Data collection and processing

The remaining 30 provinces cover the geographical locations of China's eastern, central, and western regions, including economically developed provinces such as Beijing and Shanghai and underdeveloped provinces like Qinghai and Gansu, representing various sample types. The sample data mainly comes from the following aspects. The data on the antecedent conditions for technological innovation were sourced from the China Patent Star Search System. Information regarding technological talent was extracted from the 2022 China Science and Technology Statistical Yearbook. The official websites of the provincial governments and the 2022 China Statistical Yearbook were consulted to gather data on the government's financial support and promotional efforts for the innovative concept of digitalization and low carbonization in the construction industry. The data on environmental regulation were sourced from the 2022 China Energy Statistics Yearbook and the 2022 China Statistical Yearbook. The innovation efficiency of digitalization and low carbonization integration in the construction industry was obtained from the 2022 China Energy Statistics Yearbook and the 2022 China Statistical Yearbook.

## Outcome variables

Integrating digitalization and low carbonization innovations can enhance industrial efficiency and synergy for carbon reduction, ultimately reflected in the industry's total carbon emissions and carbon emission intensity [45]. This paper refers to the research by Gong Xiaochen [46], which uses the construction industry unit GDP carbon emission index. Carbon emission intensity is used to measure of the digitalization and low carbonization integration of innovation efficiency indicators within the construction sector. This choice is due not only to the numerical size of the unit GDP carbon emission index but also to the negative correlation between the level of digitalization and low carbonization integration and innovation efficiency in the construction industry. In order to adapt to the operational regulations of the fsQCA software, this paper uses the negative value of the ratio of carbon emissions in the construction industry to the total output value as a measure of the efficiency of integration of digitalization and low carbonization innovation. Within the construction industry, a higher ratio indicates greater innovation efficiency and a lower ratio indicates lower innovation efficiency.

The method used to measure CO2 emissions involves collating the end-use energy consumption of the construction industry in each province during 2021. The carbon emission calculation formula is based on the research by Gong Xiaochen [46]. It follows the National Greenhouse Gas Inventory Guidelines set by the IPCC of the United Nations Intergovernmental Panel on Climate Change [47] and the GB/T51366-2019 'Standard for Calculating Carbon Emissions from Buildings'. The formula for calculating carbon emissions is as follows.

$$CO_2 = \sum_{i=1}^{14} CO_{2,i} = \sum_{i=1}^{14} E_i \times NCV_i \times CC_i \times COF_i \times \frac{44}{12} \qquad (1)$$

In this formula, CO2 represents the total carbon dioxide emissions from various types of energy consumption. The variable (i) represents 14 types of energy fuels (such as raw coal, paraffin, petrol, diesel, petroleum asphalt, liquefied petroleum gas, and natural gas). (E_i) denotes the energy consumption of various building industries. (NCV_i) is the average low-level heat generation of various energy sources. (CC_i) is the carbon content per unit calorific value of these energy sources, and (COF_i) is the carbon oxidation rate. The factor $\frac{44}{12}$ represents the relative molecular mass ratio of carbon dioxide to carbon. The data sources include the 2022 China Energy Statistics Yearbook and Table A.0.1 in Appendix A of the GB/T51366-2019 'Standard for Calculating Carbon Emissions from Buildings'.

## Antecedent conditions

**Technical conditions.** The level of technological innovation is measured by the total number of patent applications in the construction industry across various provinces, following the common practices in existing research. This study manually inputs "construction industry" as a keyword into the China Patent Star search system to retrieve data, compiling statistics by provincial administrative regions. This serves as the data source for technological innovation levels. Regarding technical talents, we refer to the research by Lin Yan [48] and use the number of full-time equivalent research and experimental development personnel in each province in 2021 as our data source.

**Organizational conditions.** The government's financial support is assessed based on the methodology of Yao Fengge and Wang Tianhang [49]. Since the Housing and Urban-Rural Development Committees in each province manage their respective construction industries, the ratio of the annual financial allocation from these committees in 2021 to the GDP of the construction industry for the same year is used to measure the strength of government financial support in each region. A higher ratio indicates stronger support, while a lower ratio indicates weaker support. Zhou Ling's research informs the measurement of government promotion and publicity [50]. It uses the adoption of policies by provincial governments for the coordinated development of digitalization and low carbonization in the construction industry as a benchmark. This is quantified by counting the information released or forwarded by provincial government official websites from 2020 to 2022 related to activities such as training sessions, competitions, and observation meetings concerning the coordinated development of digitalization and low carbonization in the construction industry.

**Environmental conditions.** Currently, there are numerous methods to measure environmental regulations. These include using the ratio of environmental or industrial pollution investment to an enterprise's primary business or industrial added value or evaluating compliance with industrial wastewater discharge standards. This article adopts the methodology proposed by Han Xianfeng [51], utilizing the ratio of GDP in the construction industry to the total energy consumption of the construction industry as an indicator of the intensity of environmental regulation. A larger ratio signifies stronger environmental regulation intensity, while a smaller ratio indicates weaker regulation. Table 1 presents the variable indicator measurements and their respective data sources.

## Variable calibration

Once the raw data have been collected, a calibration process is required. In fsQCA, each condition and outcome is treated as an independent set, and each case is assigned a membership score within these sets. The process of assigning membership scores to each case is known as calibration [52]. Based on existing research [13], this study calibrated five antecedent conditions and one outcome variable—innovation efficiency in integrating digitalization

**Table 1. Description of variable assignment and data sources.**

| Category of variable | Primary Indicator | Secondary indicator | Description of assignment | Data Source |
|---|---|---|---|---|
| Antecedent Conditions | Technological conditions | Technological Innovation Level | Total number of patent applications in the construction industry by province | China Patent Star Search System |
| | | Technological Talent | Full-time equivalent of research and experimental development personnel (person-years), 2021 | China Science and Technology Statistical Yearbook 2022 |
| | Organizational Conditions | Government Financial Support | Ratio of the amount of annual government financial allocations for the Housing and Urban-Rural Development Commission in 2021 to the GDP of the construction industry in 2021 | Provincial government websites, 2022 China Statistical Yearbook |
| | | Government publicity and promotion | Number of information on training, competitions, observation sessions and other activities on the synergistic development of digitization and low carbonization of the construction industry introduced or forwarded by the official websites of provincial governments (articles), 2020–2022 | Provincial government websites |
| | Environmental conditions | Environmental regulation | Ratio of construction GDP in 2021 to total energy consumption in construction in 2021 | China Energy Statistics Yearbook 2022, China Statistics Yearbook 2022 |
| Outcome variable | Efficiency of integration of digitalization and low carbonization innovation in the construction industry | | The opposite of the ratio of carbon emissions from the construction industry to the total output value of the construction industry in 2021 adopted by each province in 2021 | China Energy Statistics Yearbook 2022, China Statistics Yearbook 2022 |

Note, the statistics in the 2022 Statistical Yearbook pertain to 2021.

and low carbonization in the construction industry—using the direct calibration method. The upper quartile (75%), median, and lower quartile (25%) of the descriptive statistics of the case samples are utilized as the three calibration anchors for full membership, crossover, and full non-membership, respectively. These details are presented in Table 2.

## Data analysis and empirical results

### Necessity analysis of individual conditions

According to the QCA research steps, analyzing the necessity of individual antecedent conditions is essential before conducting a sufficiency analysis of configurations. Therefore, this article first examines whether there are necessary conditions for achieving innovation efficiency in integrating digitalization and low carbonization within the construction industry.

**Table 2. Anchors for variable calibration.**

| Category of variable | Primary Indicator | Secondary indicator | Calibration Anchor Points | | |
|---|---|---|---|---|---|
| | | | Fully Affiliated Points | Intersection Points | Completely unaffiliated point |
| Antecedent Conditions | Technological conditions | Technological Innovation Level (CX) | 79.000 | 43.500 | 17.75 |
| | | Technological Talent (RC) | 23.532 | 12.412 | 4.725 |
| | Organizational Conditions | Government Financial Support (CZ) | 0.114 | 0.030 | 0.019 |
| | | Government publicity and promotion (TG) | 24.250 | 15.500 | 7.750 |
| | Environmental conditions | Environmental regulation (GZ) | 63.106 | 42.418 | 23.653 |
| Outcome variable | Efficiency of integration of digitalization and low carbonization innovation in the construction industry (RH) | | −2.461 | −3.903 | −6.918 |

The consistency index determines the presence of a necessary condition, where a consistency greater than 0.9 indicates the necessity of the condition for the result [53]. Table 3 presents the results of the necessity test for the innovation efficiency of digitalization and low carbonization integration in the construction industry. As seen in Table 3, the consistency index for "~ environmental regulation" in the antecedent conditions related to the integration innovation efficiency of digitalization and low carbonization in the non-high-rise construction industry is 0.928. This suggests that "~ environmental regulation" is necessary for the integration of innovation efficiency in the digitalization and low carbonization of the non-high-rise construction industry. Additionally, the consistency indices for the remaining antecedent conditions are all below 0.9, indicating that there are no other necessary conditions are influencing the high integration innovation efficiency of digitalization and low carbonization in the construction industry.

## Sufficiency analysis of the conditional configuration

A key step in drawing research conclusions is the sufficiency analysis of conditional configurations. This step examines how different configurations, formed by multiple antecedent conditions, lead to the sufficiency of the outcomes. Drawing on the work of Du Yunzhou [13], this paper sets the frequency threshold at 1 and the original consistency threshold at 0.8. When the consistency index exceeds 0.8, it is considered that various configuration paths can sufficiently produce the outcomes. The core conditions in the configuration solution are identified by comparing the intermediate solution with the parsimonious solution [12]. The results of the conditional configuration sufficiency are detailed in Table 4 of this paper. According to Table 4, four groups (G1a, G1b, G1c, G2) produce high levels of innovation efficiency in integrating digitalization and low carbonization in the construction industry. Additionally, four groups (NG1, NG2a, NG2b, NG3) produce low levels of innovation efficiency in the same integration. Based on the conclusions by Fiss [52], groups G1a, G1b, and G1c share core conditions, indicating that they are equivalent second-order configurations. Similarly, NG2a and NG2b also share core conditions, making them equivalent second-order configurations.

Configuration G1a indicates that a core condition of high environmental regulation, combined with marginal conditions of high technological innovation levels and a high level of technical talent, can lead to high efficiency in integrating digitalization and low carbonization within the construction industry. The consistency for this configuration is 0.896, the

**Table 3. Analysis of necessity conditions.**

| Antecedent Conditions | High efficiency of integration of digitalization and low carbonization innovation in the construction industry | | Non-high efficiency of integration of digitalization and low carbonization innovation in the construction industry | |
|---|---|---|---|---|
| | Consistency indicators | Coverage indicators | Consistency indicators | Coverage indicators |
| Technological Innovation Level | 0.611 | 0.641 | 0.456 | 0.475 |
| ~ Technological Innovation Level | 0.500 | 0.481 | 0.656 | 0.626 |
| Technological Talent | 0.649 | 0.664 | 0.408 | 0.415 |
| ~ Technological Talent | 0.428 | 0.421 | 0.670 | 0.655 |
| Government Financial Support | 0.492 | 0.505 | 0.605 | 0.617 |
| ~ Government Financial Support | 0.627 | 0.615 | 0.515 | 0.502 |
| Government publicity and promotion | 0.512 | 0.532 | 0.544 | 0.560 |
| ~ Government publicity and promotion | 0.575 | 0.559 | 0.545 | 0.527 |
| Environmental regulation | 0.894 | 0.926 | 0.289 | 0.297 |
| ~ Environmental regulation | 0.321 | 0.312 | 0.928 | 0.897 |

**Table 4. Adequacy analysis of the conditional configuration.**

| Antecedent Conditions | High efficiency of integration of digitalization and low carbonization innovation in the construction industry | | | | Non-high efficiency of integration of digitalization and low carbonization innovation in the construction industry | | | |
|---|---|---|---|---|---|---|---|---|
|  | G1a | G1b | G1c | G2 | NG1 | NG2a | NG2b | NG3 |
| Technological Innovation Level (CX) | ● |  | ⊗ | ● | ● | ⊗ | ⊗ |  |
| Technological Talent (RC) | ● | ● | ⊗ | ⊗ | ⊗ | ⊗ | ⊗ | ● |
| Government Financial Support (CZ) |  | ⊗ | ● | ● |  | ● |  | ⊗ |
| Government publicity and promotio (TG) |  | ⊗ | ⊗ | ● | ⊗ |  | ● | ● |
| Environmental regulation (GZ) | ● | ● | ● | ⊗ | ⊗ | ⊗ | ⊗ | ⊗ |
| Consistency | 0.896 | 0.939 | 0.977 | 0.805 | 0.950 | 0.908 | 0.879 | 0.959 |
| Raw Coverage | 0.520 | 0.255 | 0.223 | 0.096 | 0.164 | 0.484 | 0.263 | 0.284 |
| Unique Coverage | 0.328 | 0.054 | 0.138 | 0.035 | 0.058 | 0.247 | 0.038 | 0.205 |
| Solution Consistency | 0.907 | | | | 0.916 | | | |
| Solution Coverage | 0.774 | | | | 0.817 | | | |

(Note, A. The presence of core conditions is marked by ●, and their absence is indicated by ⊗. The presence of marginal conditions is marked by ●, and their absence is indicated by ⊗.B. Given that government financial support is the main area of interest for researchers studying the level of influencing factors [38] and since governmental power still holds a significant position [13], it is recommended that in the "standard analysis" step of the "High efficiency of integration of digitalization and low carbonization innovation in the construction industry" operation, the variables "technological innovation level * government financial support * government publicity and promotion" be considered while excluding "technological innovation level * ~ technological talent * government publicity and promotion". In the "standard analysis" step of the operation, "non-high efficiency of integration of digitalization and low carbonization innovation in the construction industry", "~Government financial support*environmental regulation" was selected, while "technological talent*environmental regulation" was discarded.)

unique coverage is 0.328, and the raw coverage is 0.520. This implies that this path accounts for approximately 52% of the cases of high efficiency in integrating digitalization and low carbonization in the construction industry. In comparison, this path can exclusively explain 32.8% of the cases. Configuration G1a suggests that a construction ecosystem characterized by strong environmental regulations, high technological innovation, and a highly skilled workforce can lead to efficient integration of digitalization and low carbonization innovations. It also indicates that the influence of organizational conditions on achieving high efficiency in this area is relatively insignificant. Typical cities is this category include Beijing, Jiangsu, Zhejiang, Hubei, Shanghai, Guangdong, Shandong, Anhui, Hebei, and Shaanxi, as detailed in Fig 2.

Taking Beijing as an example, the city places significant emphasis on the innovative development of integrating digitalization and low carbonization within the construction industry. During the 14th Five-Year Plan period, Beijing implemented a range of standards, including the "Building Information Modeling and Engineering Acceptance Data Interaction Standards," "Technical Standards for Energy-Saving Doors and Windows in Civil Buildings," "Technical Regulations for Energy-Saving and Green Renovation of Existing Public Buildings," "Technical Regulations for Smart Community Construction," and "Smart Community Evaluation Standards." In 2021 alone, the city issued six "Green Building Evaluation Standards," 22 "Prefabricated Building Evaluation Standards," and two "Technical Standards for Energy-Saving Doors and Windows in Civil Buildings," thereby establishing a relatively comprehensive standard system to promote the integrated innovative development of digitalization and low carbonization in the construction industry.

Moreover, Beijing's construction industry accounts for 8.91% of the total number of construction industry patents across 30 case provinces, ranking fourth. The number of technical talents reaches 338,300 person-years, placing the city fifth among all provinces. Beijing exemplifies the typical characteristics of high environmental regulation, advanced technological innovation, and a robust pool of high-tech talents. Together, these factors enhance the

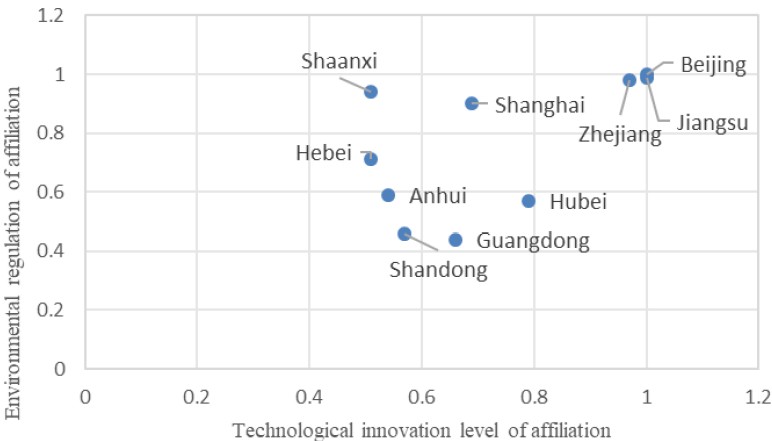

**Fig 2. Typical provinces with a type G1a configuration.**

efficiency of integrated innovation in the digitalization and low carbonization of the construction industry.

Configuration G1b indicates that a combination of core conditions—high environmental regulation—and marginal conditions such as abundant technical talent, limited government financial support, and minimal government promotion can lead to high efficiency in integrating digitalization and low carbonization innovation within the construction industry. The consistency of this configuration is 0.939, the unique coverage is 0.054, and the raw coverage is 0.255. This suggests that this path accounts for approximately 25.5% of the cases of high efficiency in integrating digitalization and low carbonization in the construction industry, with only 5.4% being solely explained by this path.

Configuration G1b implies that it is crucial to intensify environmental regulation in regions where government financial support is lacking and promotional efforts are weak. Simultaneously, these regions should focus on cultivating technical talent and introducing preferential policies to attract skilled professionals. This strategy aims to attract high-quality, multidisciplinary talent from both domestic and international sources to enhance efficiency in the integration and innovation of digitalization and low carbonization within the local construction industry, ultimately achieving the "dual carbon" goals. Typical cities in this category include Fujian, Zhejiang, and Hebei, as shown in Fig 3. Taking Hebei Province as an example, it has issued a series of standards such as the "Prefabricated Evaluation Standards," "Ultra-Low Energy Consumption Public Building Energy Saving Design Standards," "Green Building Evaluation Standards," and "Building Information Modeling Project Implementation Regulations" to bolster environmental regulations.

Regarding technical talent, Hebei Province adopts a dual approach to attract and retain talent. It not only introduces high-end talent by establishing "senior talent recruitment vouchers," providing employment and housing subsidies, implementing a "ranking and leadership" system, and holding regular innovation and entrepreneurship competitions but also cultivates composite technical talent by fostering "double first-class" universities, setting up municipal-level industrial technology research institutes, providing financial support for postdoctoral work, and establishing public training bases for talent. The example of Hebei Province illustrates how, under limited organizational capacity, integrating environmental and technological drivers can achieve high efficiency in the construction industry's digitalization and low carbonization innovation.

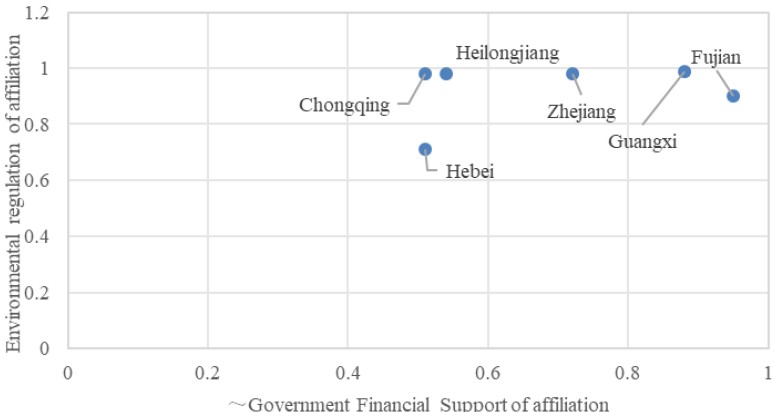

**Fig 3. Typical provinces in the G1b, G1c and G2 configurations.**

Configuration G1c suggests that efficient integration and innovation for digitalization and low carbonization within the construction industry can be achieved through a combination of high environmental regulation and certain marginal conditions: non-high technological innovation levels, limited technical talent, substantial government financial support, and limited government publicity and promotion. The consistency of this configuration is 0.977, with a unique coverage of 0.138 and a raw coverage of 0.223. This indicates that this path accounts for approximately 22.3% of the cases of high efficiency in integrating digitalization and low carbonization in the construction industry, with 13.8% of such cases being solely explained by this path. Configuration G1c implies that in regions where there is a lack of technical talent, low technological levels, and insufficient government promotion, it is still possible to achieve high efficiency in the integration of digitalization and low carbonization in the construction industry by strengthening environmental regulations and increasing government financial support. Typical cities within this category include Chongqing and Guangxi, as detailed in Fig 3. Taking Chongqing as an example, in 2021, the city introduced a series of standards to promote the high-quality and high-standard development of green buildings. These include the "Technical Guidelines for Green Building Evaluation in Chongqing," "Technical Guidelines for the Construction of Green Ecological Residential (Green Building) Communities," "Technical Review Guidelines for Energy-saving (Green Building) Design of Public Buildings," and "Technical Review Guidelines for Energy-saving 65% (Green Building) Design." These documents aim to guide the design, review, evaluation, and management of green buildings.

Configurations G1a, G1b, and G1c collectively demonstrate the positive impact of the "Porter Hypothesis," indicating that environmental regulations significantly enhance the efficiency of digital and low carbonization integrated innovation in the construction industry.

Configuration G2 reveals that a combination of high technological innovation levels, substantial government financial support, and low environmental regulation, along with marginal conditions such as limited technical talent and strong government promotion, can lead to high efficiency in integrating digitalization and low carbonization within the construction industry. The consistency of this configuration is 0.805, with a unique coverage of 0.035 and a raw coverage of 0.096. This indicates that this path accounts for approximately 9.6% of the cases of high efficiency in integrating digitalization and low carbonization in the construction industry, with 3.5% being solely explained by this path.

Configuration G2 suggests that in settings where environmental regulation is weak and technical talent is lacking, efforts to elevate technological innovation levels, increase

government financial support, and intensify government promotion of digitalization and low carbonization integration concepts can enhance the integration efficiency.

## Pathways to the emergence of innovative efficiency in the integration of digitalization and low carbonization in the non-high construction industry

In order to study the driving mechanisms behind the digitalization and low carbonization integration innovation efficiency in the non-high-rise construction industry, this article also examines the configurational paths that lead to the efficiency of digitalization and low carbonization integration innovation in this sector. From Table 4, it can be seen that there are four configuration paths: Configuration NG1, Configuration NG2a, Configuration NG2b, and Configuration NG3. Configuration NG1 indicates that when the region is under non-high environmental regulation, non-high government promotion, and non-high technical talent, even if there is a high level of technological innovation, it will still result in non-high efficiency of the integration of digitalization and low carbonization development in the construction industry. At this point, the level of government financial support does not affect the outcome. Configurations NG2a and NG2b indicate that under conditions of non-high technological innovation, non-high environmental regulation, and non-high technical talent, even with high government financial support or high government promotion, the efficiency of the integration of digitalization and low carbonization development in the construction industry will not be high. Configuration NG3 indicates that when the region is under non-high environmental regulation and non-high government financial support, even with high technical talent and government promotion, there will not be high efficiency in integrating digitalization and low carbonization development in the construction industry. Configurations NG1, NG2a, NG2b, and NG3 demonstrate that non-high environmental regulation plays a key role in the outcomes of non-high efficiency of integrating digitalization and low carbonization development in the construction industry.

## Robustness test

This article references the research of Chen Huaichao et al. [54]. It conducts a robustness test on the configurational antecedents of the integration innovation efficiency of high-rise construction digitalization and low carbonization from two aspects: adjusting the frequency from 1 to 2 and the original consistency threshold from 0.8 to 0.85. The experimental results are shown in Tables 5 and 6. From the tables, it can be seen that configuration G1a' is a subset of configuration G1a, and the results of configuration G1c' are consistent with those of configuration G1c; configurations G1a", G1b", and G1c" are consistent with the results of configurations G1a, G1b, and G1c, respectively. This indicates that the results of the study are robust.

## Analysis of the differentiated roadmap for the integration of digitalization and low carbonization innovation efficiency in the construction industry in East, Central and West regions

Given the differences in geographical location, resource distribution, levels of economic development, and policy frameworks among various provinces, these factors may influence the levels of technological innovation, technical talent, government financial support, and environmental regulations to varying degrees. This, in turn, affects the efficiency of the construction industry in each province regarding the integration of digitalization and low carbonization innovation. Therefore, this study follows the resolutions of the 16th National People's Congress Fourth Session and the 18th National People's Congress Fifth Session, categorizing the case provinces

Table 5. Robustness test of increasing frequency.

| Antecedent Conditions | High efficiency of integration of digitalization and low carbonization innovation in the construction industry(Frequency 1) | | | | High efficiency of integration of digitalization and low carbonization innovation in the construction industry(-Frequency 2) | |
|---|---|---|---|---|---|---|
| | G1a | G1b | G1c | G2 | G1a′ | G1c′ |
| Technological Innovation Level (CX) | ● | | ⊗ | ● | ● | ⊗ |
| Technological Talent (RC) | ● | ● | ⊗ | ⊗ | ● | ⊗ |
| Government Financial Support (CZ) | | ⊗ | ● | ● | ⊗ | ● |
| Government publicity and promotion (TG) | | ⊗ | ⊗ | ● | | ⊗ |
| Environmental regulation (GZ) | ● | ● | ● | ⊗ | ● | ● |
| Consistency | 0.896 | 0.939 | 0.977 | 0.805 | 0.881 | 0.977 |
| Raw Coverage | 0.520 | 0.255 | 0.223 | 0.096 | 0.423 | 0.223 |
| Unique Coverage | 0.328 | 0.054 | 0.138 | 0.035 | 0.375 | 0.175 |
| Solution Consistency | 0.907 | | | | 0.909 | |
| Solution Coverage | 0.774 | | | | 0.598 | |

(Note, A. The presence of core conditions is marked by ●, and their absence is indicated by ⊗. The presence of marginal conditions is marked by ●, and their absence is indicated by ⊗.)

Table 6. Robustness test for raising the original conformance threshold.

| Antecedent Conditions | High efficiency of integration of digitalization and low carbonization innovation in the construction industry (Original consistency threshold 0.8) | | | | High efficiency of integration of digitalization and low carbonization innovation in the construction industry (Original consistency threshold 0.85) | | |
|---|---|---|---|---|---|---|---|
| | G1a | G1b | G1c | G2 | G1a″ | G1b″ | G1c″ |
| Technological Innovation Level (CX) | ● | | ⊗ | ● | ● | | ⊗ |
| Technological Talent (RC) | ● | ● | ⊗ | ⊗ | ● | ● | ⊗ |
| Government Financial Support (CZ) | | ⊗ | ● | ● | | ⊗ | ● |
| Government publicity and promotion (TG) | | ⊗ | ⊗ | ● | | ⊗ | ⊗ |
| Environmental regulation (GZ) | ● | ● | ● | ⊗ | ● | ● | ● |
| Consistency | 0.896 | 0.939 | 0.977 | 0.805 | 0.895 | 0.939 | 0.977 |
| Raw Coverage | 0.520 | 0.255 | 0.223 | 0.096 | 0.520 | 0.255 | 0.223 |
| Unique Coverage | 0.328 | 0.054 | 0.138 | 0.035 | 0.336 | 0.054 | 0.138 |
| Solution Consistency | 0.907 | | | | 0.920 | | |
| Solution Coverage | 0.774 | | | | 0.739 | | |

(Note, A. The presence of core conditions is marked by ●, and their absence is indicated by ⊗. The presence of marginal conditions is marked by ●, and their absence is indicated by ⊗.)

into three major regions: eastern, central, and western. By conducting a comparative study of the construction industry's performance in these regions concerning the efficiency of digitalization and low carbonization integration innovation, we aim to explore how technological, organizational, and environmental factors influence the efficiency of the construction industry in different regions through various configurations. The experimental results are shown in Table 7.

Based on Table 7, an evident configuration path is observed in the eastern region. Configuration Path 1 suggests that a high level of technological innovation, abundant technical talent, and robust government initiatives in environmental regulation can sufficiently drive

**Table 7. Grouping analysis of innovation efficiency in the integration of digitalization and low carbonization in the high construction industry in the East, Central and West regions.**

| Antecedent Conditions | Eastern Region | Central Region | | | Western Region | |
|---|---|---|---|---|---|---|
| | Configuration 1 | Configuration 2 | Configuration 3 | Configuration 4 | Configuration 5 | Configuration 6 |
| Technological Innovation Level (CX) | ● | ⊗ | ⊗ | ● | ● | ● |
| Technological Talent (RC) | ● | ● | ⊗ | ⊗ | ● | ● |
| Government Financial Support (CZ) | | ⊗ | ● | ● | ● | ⊗ |
| Government publicity and promotion (TG) | ● | ⊗ | ⊗ | ● | ⊗ | ● |
| Environmental regulation (GZ) | ● | ● | ● | ⊗ | ● | ● |
| Consistency | 0.947 | 0.921 | 1.000 | 0.924 | 1.000 | 0.987 |
| Raw Coverage | 0.525 | 0.165 | 0.344 | 0.147 | 0.257 | 0.465 |
| Unique Coverage | 0.525 | 0.066 | 0.241 | 0.127 | 0.181 | 0.389 |
| Solution Consistency | 0.947 | 0.954 | | | 0.991 | |
| Solution Coverage | 0.525 | 0.537 | | | 0.646 | |

Note: Since government financial support is the most focused influencing factor for researchers [38], and government power remains a primary position [13], the "standard analysis" steps for the "central region" selected " ~ TGfz*GZfz" and "CXfz*CZfz"; also, because environmental regulation is a core condition for multiple pathways, the "standard analysis" steps for the "western region" selected "GZfz".

the integration and innovative efficiency of digitalization and low carbonization in the construction industry, even without direct financial support from the government. The eastern region's superior geographical advantages, foundational historical development, and early focus on reform and opening-up policy initiatives have contributed to an elevated economic development level compared to the central and western regions. This economic strength not only cultivates an environment conducive to technological innovation and significant advancements in new technology research and application but also draws a substantial influx of technical talent nationally and internationally, creating a rich talent pool. With the rapid economic growth in the eastern region, the quality of life for its residents has significantly improved, leading to a heightened public awareness and concern for environmental protection and sustainable development. This growing social consciousness provides a solid base of public support for governmental efforts in promoting ecological civilization. Consequently, in response to the public's growing preference for a green lifestyle, the government in the eastern region is likely to intensify the promotion of policies aimed at advancing digital low-carbon development in the construction industry. Moreover, stricter environmental regulations are being implemented to guide and control the coordinated development of digitalization and low carbonization within the industry.

Analysis of the three configurational paths in the central region reveals distinct challenges and opportunities for integrating digitalization and low carbonization within the construction industry. Configurations 2 and 3 highlights that when the levels of technological innovation are low, government financial support is inadequate, or technical talent is scarce, the lack of strong governmental promotion and advocacy limits the efficiency of integrating digitalization with low carbonization. In these scenarios, enhancing environmental regulations emerges as a crucial condition, while bolstering technical talent reserves or increasing government financial support are supplementary measures to facilitate this integration.

Configuration 4, on the other hand, indicates that in the absence of sufficient technical talent and stringent environmental regulations, the advancement of digitalization and low carbonization in the construction sector can be effectively driven by elevating technological innovation levels, boosting government financial support, and enhancing the promotion of relevant concepts. The central region's proximity to the developed provinces in the eastern

region affords it certain strategic advantages. According to the experimental results, the central region can achieve high efficiency in integrating digitalization and low carbonization within the construction industry by having robust environmental regulations, advanced technological innovation, and substantial government financial support as pivotal elements. These core conditions can interact with various additional conditions to significantly influence the integration process.

Compared to the eastern and central regions, the western region has a lower economic level and faces certain disadvantages regarding technology, organization, and environment. Improving the efficiency of integrating digitalization and low carbonization innovation in the construction industry can enhance technological innovation levels, cultivate technical talent, strengthen environmental regulations, increase government financial support, or boost government promotion efforts. In summary, China's eastern, central, and western regions exhibit different driving models in promoting the efficiency of integrating digitalization and low carbonization innovation in the construction industry. These regions show significant regional differences in economic development levels due to variations in industrial structure, historical background, regional culture, and degrees of openness. This leads to unique challenges and opportunities in advancing the integration of digitalization and low carbonization innovation in the construction industry.

## Conclusions and limitations

### Conclusions of the study

This article is based on the TOE theoretical framework. It uses the fsQCA method to explore the driving mechanisms of the integration of digitalization and low carbonization innovation efficiency in the construction industry from a configurational perspective, using data from 30 provinces (municipalities and autonomous regions) as sample data. The research results are as follows:

(1) From a holistic perspective, "environmental regulation" is a necessary condition for the integration and innovative efficiency of digitalization and low carbonization in the non-high-rise construction industry, and there is no necessary condition affecting the innovative efficiency of digitalization and low carbonization integration in the high-rise construction industry. This indicates that, without effective environmental regulation, the integration and innovative development of digitalization and low carbonization in the construction industry may be subject to certain constraints.

(2) In integrating digitalization and low carbonization innovation efficiency in the construction industry, three-quarters of the group paths include high environmental regulations. Additionally, 100% of the configurations in non-construction industry digitalization and low-carbon integration innovation efficiency include non-high environmental regulations. This fully demonstrates the "Porter Hypothesis," which suggests that appropriate environmental regulations can stimulate corporate innovation vitality and improve production efficiency. Environmental regulations also play a key driving role in enhancing the digitalization and low carbonization integration innovation efficiency of the construction industry.

In light of the points above, the following recommendations are put forth for consideration:

Firstly, it is recommended that the supervision and management of the construction industry be strengthened. The government should implement a robust regulatory system and effective supervision mechanism to guarantee that all construction projects adhere to

environmental protection and low carbonization standards throughout the entire process, from design to construction to operation. Additionally, it should enhance the qualification audits and daily supervision of construction enterprises, impose penalties on those who violate environmental protection policies, and publicly expose them to create a deterrent effect. Secondly, it is imperative to establish rigorous environmental policies and standards. It is recommended that stricter construction environmental regulations and standards, such as energy efficiency standards and carbon emission limits, be introduced to legally enforce enterprises to enhance their environmental awareness. Furthermore, policy incentives, such as tax incentives and capital subsidies, should be employed to encourage enterprises to increase investment in environmental protection technology and equipment. Thirdly, financial support and the promotion of exemplary cases. It is recommended that a special fund or guide fund be established to provide financial support for construction industry enterprises engaged in technology research and development and the application of digitalization and low carbonization. Furthermore, demonstration projects should be established to showcase the integration of digitalization and low-carbonization innovation and inspire more enterprises to innovate by promoting successful cases.

(3) The research also found that four equivalent configurational paths can drive the realization of high efficiency in integrating digitalization and low carbonization in the construction industry. This means that different provinces and cities can choose paths more suitable for the coordinated development of the "two transformations" in the construction industry based on their own conditions and development status, ultimately achieving the goal of "dual carbon." This development strategy, tailored to local conditions, considers the actual situations in various regions and provides more possibilities for improving the efficiency of digitalization and low carbonization integration innovation in the construction industry. Therefore, local governments should formulate corresponding development strategies and policies based on local realities to enhance the efficiency of digitalization and low carbonization integration innovation in the construction industry. For instance, local governments should be permitted to conduct comprehensive research and assessment of the current state of the local construction industry, to gain insight into the extent of digitization and low carbonization within the industry and identify existing constraints and challenges. Based on the research findings, development strategies and policies should align with the local context. This may entail the provision of tax incentives and capital subsidies to encourage construction enterprises to increase their investment in digitisation and low carbonization. Furthermore, the government may establish a collaborative platform for digitisation and low carbonization in the construction industry, facilitating exchanges and cooperation among enterprises, scientific research institutes, universities, and other relevant stakeholders. This platform could serve as a conduit for joint efforts to advance technological innovation and industrial upgrading. Concurrently, construction enterprises must proactively identify a development trajectory that aligns with their specific circumstances and growth aspirations. For instance, they should augment their investment in research and development about digital and low carbonization technologies, including the implementation of BIM (Building Information Model), Internet of Things, big data, and other related technologies, as well as the advancement of novel low carbonization building materials and construction methodologies. Furthermore, they should optimise their business processes for design, construction, operation, and maintenance through digital means to enhance efficiency and reduce costs. Additionally, they should proactively cultivate collaborative relationships with governmental entities, scientific research institutions, academic institutions, and other relevant parties. To this

end, we will proactively engage with government entities, research institutions, academic institutions, and other relevant parties to foster technological innovation and talent cultivation. Furthermore, we will implement incentive structures to motivate employees to contribute to technological advancement and industrial transformation actively.

## Theoretical significance

Firstly, when analysing the complex issue of digital transformation and low carbonization development of the construction industry, this paper employs the TOE theoretical framework in a clever and integrated manner, constructing a comprehensive and in-depth research framework to systematically explore the key factors affecting the efficiency of the integration of digitalization and low carbonization innovation in the construction industry. The TOE theoretical framework originally emphasises the impact of technological factors, internal organizational factors and external environmental factors on corporate strategy and performance. The TOE theoretical framework originally emphasises the influence of technological factors, internal organisational factors, and external environmental factors on corporate strategy and performance. This paper builds on this by incorporating the construction industry's unique industry characteristics, policy orientation, and market demand, thus forming a set of analyses that is both universal and industry-specific. This research framework offers academics and practitioners a systematic and comprehensive understanding of the multi-dimensional influencing factors of the innovation efficiency of integrating digitalization and low carbonization in the construction industry. Furthermore, it provides novel research perspectives and theoretical support for subsequent scholars' research in this field, thereby facilitating the deepening and expansion of academic research.

Secondly, a significant methodological contribution of this study is the introduction of the group perspective, which represents a noteworthy theoretical advancement in the study of the factors influencing the efficiency of integrating digitalization and low carbonization in the construction industry. This perspective allows the study to analyze not only the direct effect of a single factor on the convergence innovation efficiency but, more importantly, to reveal the multiple path effects of multiple factors in different configurations and combinations. This provides a powerful analytical tool to address the complexity and uncertainty inherent in the construction industry's convergence of digitization and low carbonization. Furthermore, this research methodology offers practical guidance on how the construction industry can enhance the innovation efficiency of digitalization and low carbonization integration. It encourages enterprises and organizations in the industry to consider not only introducing a single technology or implementing a policy but also the synergy between factors as a whole and the combination of paths most suitable for their development when promoting change. This provides new perspectives and a solid foundation for the realizing a green, smart and sustainable construction industry. The objective is to provide new perspectives and a solid theoretical basis for the realizing green, intelligent and sustainable development in the construction industry.

## Shortcomings and prospects

(1) Due to the limited research and insufficient statistical data on the integration and innovation of digitalization and low carbonization in the construction industry, this article only uses static data from 2021. Future researchers can collect dynamic data from the past 5 to 10 years and analyze it using the QCA method.

(2) In this study, the fsQCA research method was employed. The calibration process of fsQCA is highly sensitive, and different calibration methods and parameter settings may

yield disparate analysis results, thereby increasing the complexity and unpredictability of the findings. In contrast, NCA can provide a more stable datum point for the analysis by identifying the necessary conditions. It is recommended that in the future, scholars employ the fsQCA method in conjunction with the NCA method to mitigate the uncertainty inherent in fsQCA analyses by leveraging the stability of NCA to enhance the precision and dependability of the outcomes.

(3) This article summarizes the experiences and research of predecessors and selects five important aspects as antecedent conditions. Future research can be conducted on integrating digitalization and low carbonization innovation efficiency in the construction industry by adding or changing these antecedent conditions.

## Supporting information

**S1 Data.** **Summary of data in Figures 2 and 3**. This is the Figures 2 and 3 title. (DOCX)

**S2 Data.** **Summary of data in Table 4–6**. This is the Table 4–6 title. (DOCX)

**S3 Data.** **Summary of data in Table 7**. This is the Table 7, title. (DOCX)

## Acknowledgments

We are grateful to all of those who provided useful suggestions for this study.

## Author contributions

**Conceptualization:** Shiming Wang.

**Data curation:** Haifeng Xiong.

**Formal analysis:** Haifeng Xiong.

**Funding acquisition:** Shiming Wang.

**Investigation:** Haifeng Xiong.

**Methodology:** Haifeng Xiong.

**Project administration:** Shiming Wang.

**Resources:** Shiming Wang.

**Software:** Haifeng Xiong.

**Supervision:** Shiming Wang.

**Writing – original draft:** Haifeng Xiong.

**Writing – review & editing:** Haifeng Xiong.

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
