## [Decision Letter · Decision Letter 0]

15 Oct 2024

PONE-D-24-40808What influences the innovation efficiency of digitalization and low carbonization integration in the construction industry? -Configuration analysis based on fsQCAPLOS ONE

Dear Dr. Xiong,

Thank you for submitting your manuscript to PLOS ONE. After careful consideration, we feel that it has merit but does not fully meet PLOS ONE’s publication criteria as it currently stands. Therefore, we invite you to submit a revised version of the manuscript that addresses the points raised during the review process.

We look forward to receiving your revised manuscript.

Kind regards,

Chante Jian Ding

Academic Editor

PLOS ONE

**Journal Requirements:**

This work was supported in part by the Basic Scientific Research Project of EducationDepartment of Liaoning Province under grant LJKMR20220709 and in part by the Social Science Fund of Liaoning Province under grant 23-A045.  The funders had no role in study design, data collection and analysis, decision to publish, or preparation of the manuscript.

Reviewers' comments:

Reviewer's Responses to Questions

**Comments to the Author**

1. Is the manuscript technically sound, and do the data support the conclusions?

Reviewer #1: Partly

Reviewer #2: No

2. Has the statistical analysis been performed appropriately and rigorously? 

Reviewer #1: Yes

Reviewer #2: No

3. Have the authors made all data underlying the findings in their manuscript fully available?

Reviewer #1: Yes

Reviewer #2: Yes

4. Is the manuscript presented in an intelligible fashion and written in standard English?

Reviewer #1: Yes

Reviewer #2: No

5. Review Comments to the Author

**Reviewer #1: ** （1）The introduction does a good job of framing the study within the dual contexts of digitalization and the “double carbon” goal. However, the discussion on how the construction industry’s carbon emissions align with these global and national goals could be expanded. Adding specific challenges faced by the industry in integrating digitalization and low carbonization would strengthen the relevance of your research.

（2）The statement of the problem should explicitly highlight what existing literature lacks concerning the integration of digital and low-carbon innovations. Clearly stating the three key questions in bullet points would also improve readability.

（3）The current review mixes several themes, such as digitalization, low carbonization, and BIM technology. I recommend separating these into distinct subsections (e.g., digitalization in construction, low carbonization initiatives, and synergies between them). This would make the literature review easier to follow and emphasize the progression of research

（4）While many studies are mentioned, the review lacks critical engagement with these sources. You should critically assess how past research either supports or contradicts your findings.

（5）It would be helpful to provide a more detailed explanation of why fsQCA is better suited to this study than other quantitative methods. How does it handle the complexity of multiple interacting factors better than regression-based models?

（6）it would be helpful to deepen the theoretical implications of your findings. You could also discuss how your findings challenge or extend existing theories in digital and low-carbon innovation.

（7）A thorough proofreading for grammar and consistency will improve the overall quality of the manuscript.

**Reviewer #2: ** This paper examines the factors influencing the innovation efficiency of digitalization and low carbonization integration in the construction industry across 30 Chinese provinces in 2021, using a configuration analysis based on fuzzy-set Qualitative Comparative Analysis (fsQCA). The paper contains several analytical flaws that need to be addressed. I recommend a major revision of the manuscript.

Comments and Suggestions

1. This study uses cross-sectional data from 30 provinces, which is a relatively small sample size. How do the authors ensure the representativeness of the selected sample? The authors should discuss potential biases arising from sample selection.

2. Although the fsQCA method can identify combinations of conditions, it is difficult to establish strict causal relationships. However, judging from the paper's title, the authors clearly intend to discuss the factors influencing the innovation efficiency of digitalization and low carbonization integration in the construction industry.

3. The use of static cross-sectional data overlooks the dynamic characteristics of innovation efficiency over time. Why didn't the paper consider panel data over multiple years?

4. The paper lacks heterogeneity analysis of differences between provinces.

5. The policy recommendations based on the research results are relatively general and lack specific, actionable measures.

6. The discussion on the limitations of the fsQCA method itself is insufficient, such as the sensitivity of the variable calibration process.

7. There are grammatical issues, for example, in the paper title "What influences the innovation efficiency of digitalization and low carbonization integration in the construction industry? -Configuration analysis based on fsQCA", an article "a" should be added before "Configuration analysis".

8. Some figures and tables in the paper are not standardized. For instance, Figure 2 does not explain what the x and y axes represent.

6. PLOS authors have the option to publish the peer review history of their article (what does this mean? ). If published, this will include your full peer review and any attached files.

**Do you want your identity to be public for this peer review?** For information about this choice, including consent withdrawal, please see our Privacy Policy .

Reviewer #1: No

Reviewer #2: No

---

## [Author Response · Author response to Decision Letter 1]

14 Nov 2024

Dear Experts and Teachers

Thank you for your decision and constructive feedback on our manuscript "What factors influence innovation efficiency in integrating digitalization and low carbonization within the construction industry? - A configuration analysis based on fsQCA" (ID:PONE-D-24-40808).This will certainly help us improve the quality of our manuscript.We carefully considered the suggestions of the reviewers and editors and responded item by item. However, due to the content of the reply, please refer to the attachment "Response to Reviewers" for details.

Wishing you a smooth workday and a fulfilling life!

---

## [Decision Letter · Decision Letter 1]

9 Dec 2024

What factors influence innovation efficiency in integrating digitalization and low carbonization within the construction industry? - A configuration analysis based on fsQCA

PONE-D-24-40808R1

Dear Dr. Xiong,

We’re pleased to inform you that your manuscript has been judged scientifically suitable for publication and will be formally accepted for publication once it meets all outstanding technical requirements.

Kind regards,

Sara Eloy, Ph.D

Academic Editor

PLOS ONE

Additional Editor Comments (optional):

Reviewers' comments:

Reviewer's Responses to Questions

**Comments to the Author**

1. If the authors have adequately addressed your comments raised in a previous round of review and you feel that this manuscript is now acceptable for publication, you may indicate that here to bypass the “Comments to the Author” section, enter your conflict of interest statement in the “Confidential to Editor” section, and submit your "Accept" recommendation.

Reviewer #1: All comments have been addressed

Reviewer #2: All comments have been addressed

2. Is the manuscript technically sound, and do the data support the conclusions?

Reviewer #1: Yes

Reviewer #2: Yes

3. Has the statistical analysis been performed appropriately and rigorously? 

Reviewer #1: Yes

Reviewer #2: Yes

4. Have the authors made all data underlying the findings in their manuscript fully available?

Reviewer #1: No

Reviewer #2: Yes

5. Is the manuscript presented in an intelligible fashion and written in standard English?

Reviewer #1: (No Response)

Reviewer #2: Yes

6. Review Comments to the Author

Reviewer #1: (No Response)

Reviewer #2: (No Response)

7. PLOS authors have the option to publish the peer review history of their article (what does this mean? ). If published, this will include your full peer review and any attached files.

**Do you want your identity to be public for this peer review?** For information about this choice, including consent withdrawal, please see our Privacy Policy .

Reviewer #1: No

Reviewer #2: No

---

## [Editor Report · Acceptance letter]

PONE-D-24-40808R1

PLOS ONE

Dear Dr. Xiong,

I'm pleased to inform you that your manuscript has been deemed suitable for publication in PLOS ONE. Congratulations! Your manuscript is now being handed over to our production team.

Kind regards,

on behalf of

Professor Sara Eloy

Academic Editor

PLOS ONE